# Transient targeting of hypothalamic orexin neurons alleviates seizures in a mouse model of epilepsy

Han-Tao Li [1,2], Paulius Viskaitis[1], Eva Bracey[1], Daria Peleg-Raibstein [1] & Denis Burdakov [1] ✉

Lateral hypothalamic (LH) hypocretin/orexin neurons (HONs) control brain-wide electrical excitation. Abnormally high excitation produces epileptic seizures, which affect millions of people and need better treatments. HON population activity spikes from minute to minute, but the role of this in seizures is unknown. Here, we describe correlative and causal links between HON activity spikes and seizures. Applying temporally-targeted HON recordings and optogenetic silencing to a male mouse model of acute epilepsy, we found that pre-seizure HON activity predicts and controls the electrophysiology and behavioral pathology of subsequent seizures. No such links were detected for HON activity during seizures. Having thus defined the time window where HONs influence seizures, we targeted it with LH deep brain stimulation (DBS), which inhibited HON population activity, and produced seizure protection. Collectively, these results uncover a feature of brain activity linked to seizures, and demonstrate a proof-of-concept treatment that controls this feature and alleviates epilepsy.

Epilepsy is a serious global burden on public health. This chronic and debilitating neurological disorder affects people of all ages, with an estimated 50 million sufferers worldwide. Aside from recurrent seizures, epilepsy is commonly associated with comorbidities such as cognitive, emotional, and sleep disorders[1–5]. The main treatment for epilepsy is drug therapy, however, a significant proportion of patients do not respond to drugs[6,7]. Anti-seizure drugs typically target widely-expressed ion channels and neurotransmitters, causing general suppression of brain activity[8]. This has additional limitations, including the development of drug resistance, adverse and unpredictable cognitive side effects, slow onset and offset of drug action, and patient non-compliance[9–15]. For these reasons, more specific and rapid seizure control, through electrical modulation of specific brain regions with implanted electrodes (deep brain stimulation, DBS), is seen as a promising alternative[16–18]. However, anti-seizure DBS protocols that have been tried so far are seen as

insufficiently understood and validated, and optimal DBS parameters for seizure control are still lacking[19–26]. Therefore, identification of new targets and modalities for epilepsy therapy remains important.

Among neural populations that govern brain-wide electrical activity, the most recently identified are hypocretin/orexin neurons (HONs)[27–31]. HONs are found exclusively in the lateral hypothalamus, but project brain-wide, where they release several excitatory transmitters, including hypocretin/orexin peptides and glutamate[32–34]. HON activity acutely controls switching of brain electrical activity between different global states such as sleep and wakefulness[28,35], but their roles in epilepsy are not well understood. HONs are proposed to exacerbate seizures, in part due to their reciprocal connections to brain areas implicated in epileptogenesis, for example the hippocampal region, via monosynaptic and polysynaptic connections[36–45]. Thus, suppression of HON activity may

[1]Department of Health Sciences and Technology, Swiss Federal Institute of Technology | ETH Zurich, 8603 Schwerzenbach, Switzerland. [2]Section of Epilepsy, Department of Neurology, Chang Gung Memorial Hospital at Linkou Medical Center and Chang Gung University College of Medicine, 333 Taoyuan, Taiwan. ✉e-mail: denis.burdakov@hest.ethz.ch

be an attractive anti-seizure therapy. However, chronic HON suppression may have undesirable side effects on arousal and energy metabolism[46,47]. In turn, acute HON suppression is underdeveloped as therapy, because it is unknown which natural HON activity features are linked to seizures and targetable by acute DBS treatments.

Therefore, increased understanding of how natural HON activity predicts and controls seizures may open additional avenues for epilepsy treatment. This paper aims to gain this understanding through correlative and causal experiments using HON-specific neural recordings and optogenetic silencing in a mouse model of epilepsy. We also probe anti-seizure effects of lateral hypothalamic DBS temporal protocols that involve HON suppression, enabling us to better understand potential therapeutic applications.

## Results

### Pre-seizure natural HON activity correlates with seizure intensity

To produce controlled and reproducible epileptic seizures within and across subjects, we used the hippocampal optogenetic stimulation model of seizure induction[48–50] in behaving mice (Fig. 1a). By varying the frequency of hippocampal optogenetic stimulation, we found that 20 Hz, 10 s laser stimulation produced robust electrophysiological seizures (Fig. 1b, c), and a full spectrum of behavioral seizure characteristics as assessed using Racine scoring[51] (Fig. 1c). Therefore, we used this frequency to induce seizures in subsequent experiments. Systemic administration of the orexin receptor antagonist SB-334867 (30 mg/kg i.p.) suppressed seizure probability (Fig. 1d) and power (Fig. 1e). This confirms that endogenous HON activity contributed to seizures in our experimental paradigm. As expected, systemic

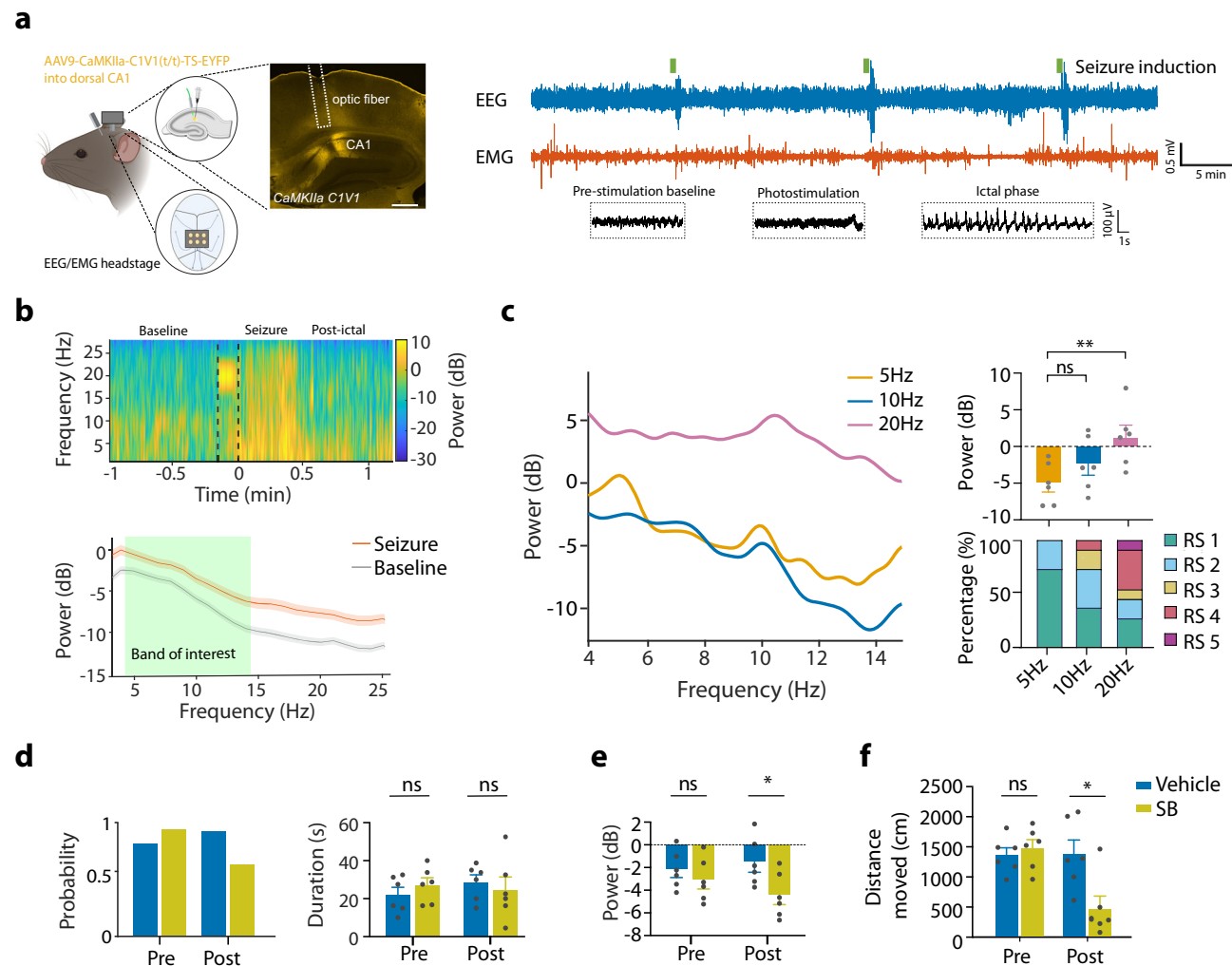

**Fig. 1 | Seizure model validation and effects of systemic blockade of hypocretin/ orexin receptors. a** Left, targeting schematic of EEG/EMG experimental paradigm (generated with BioRender), and of CaMKIIa-C1V1 expression in hippocampal CA1 for seizure induction. Scale bar is 500 μm. Right, Top, sample recordings demonstrated three bouts of seizure inductions. Bottom, EEG during pre-stimulation baseline phase, photo-stimulation, and seizure-like after-discharges during ictal phase. **b** Top, example of time-frequency heatmap of EEG power from one mouse (black dash line indicated 20 Hz laser stimulation) and Bottom, spectral power analysis of group data (mean ± SEM of $n = 6$ mice), showing 30 s before, and 30 s after seizure induction. Green highlight of the bottom panel indicated power band of interest (4–14 Hz). **c** Left, EEG power spectrum of seizure induced by 5, 10 and 20 Hz C1V1 stimulation. Right, Top, average EEG power (One-way RM ANOVA, $F_{(2,10)} = 7.114$, $P = 0.01$, Tukey's post-tests: 5 Hz vs. 20 Hz $P = 0.009$). Right, Bottom, behavioral Racine Score (RS) of seizures induced by different stimulation frequency. **d** Left, seizure probability (pre, 75 vs 83%; post, 79% vs 54% of 24 stimulation in each group) and Right, ictal duration before and 1 h after orexin antagonist SB-334867 (SB) or vehicle administration. **e** EEG average power of the induced seizures before and after SB or vehicle (two-tailed $t$ test, pre-drug $P = 0.39$, post-drug $P = 0.03$). **f** Total distance traveled of the first 10-min time bin before and after SB or vehicle (two-tailed $t$ test, pre-SB vs pre-vehicle $P = 0.54$, post-vehicle vs. post-SB $P = 0.01$). Data are mean ± SEM of $n = 6$ mice. ns = not significant, *$P < 0.05$, **$P < 0.01$. Source data are provided as a Source Data file.

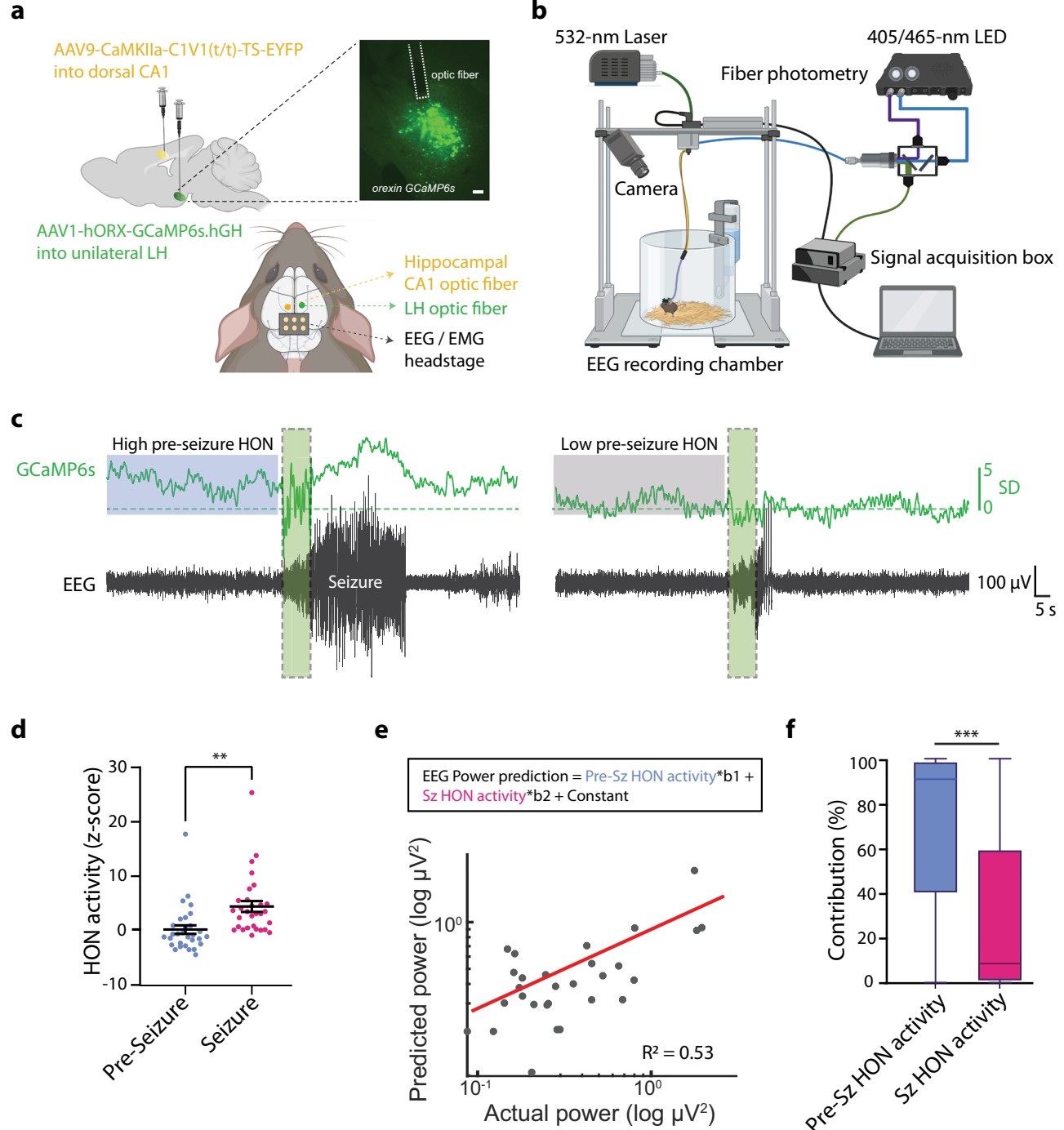

**Fig. 2 | Relationships between seizure power and different phases of peri-seizure HON activity. a** Top, Left, stereotaxic surgical schematic for virus injections, Top, Right, lateral hypothalamus coronal section showing Orx-GCaMP6s expression and fiber position in lateral hypothalamus. Scale bar is 100 μm. Bottom, relative fiber and EEG headstage locations (cartoon generated with BioRender). The figure is representative of $n = 8$ mice. **b** Schema of experimental setup for simultaneous fiber photometry, EEG recordings and seizure induction (images generated with BioRender). **c** Example EEG (black) and photometry (green) recording to demonstrate one seizure session of high (left) vs. low (right) pre-seizure HON activity. **d** Pre-seizure (blue) and during seizure (magenta) average HON activity (two-tailed $t$ test, **$P = 0.001$, $n = 30$ recordings from 8 mice). **e** General linear model to predict EEG power from measures of HON activity in different seizure phases ($R^2 = 0.53$, $n = 30$ recordings from 8 mice). **f** Relative contributions of the input prediction variables from panel **e**: center line, median; box limits, upper and lower quartiles; whiskers, 1.5x interquartile range. Two-tailed $t$ test ***$P = 0.0001$, $n = 30$. Data are mean ± SEM of $n = 8$ mice or as indicated. **d**, **e** each point represents 1 seizure event. Source data are provided as a Source Data file.

SB-334867 also had the off-target effect of reducing general locomotor activity (Fig. 1f).

In order to visualize the endogenous HON activation patterns associated with seizures, we performed LH fiberoptic recordings of GCaMP6s neural activity reporter specifically targeted to HONs[52]

(Fig. 2a, b). We observed rapid, spontaneous fluctuations in HON activity in the absence of seizures, as expected from this neural population[53] (Fig. 2c). During seizures, the HON activity robustly increased (Fig. 2c, d). However, there was no significant correlation between seizure-associated HON activity and seizure intensity

(Pearson $r = 0.2034$, $P = 0.2900$, $n = 8$ mice; correlation between mean HON activity during seizure and peak seizure intensity). In contrast, there was a significant positive correlation between seizure intensity and HON activity preceding the seizure ($P < 0.0001$, Pearson $r = 0.7119$, $n = 8$ mice). To further probe correlations between different HON activity epochs and seizure power, we used multivariate analysis featuring pre-seizure and seizure HON activity as predictors of seizure power (Fig. 2e, see Methods). This confirmed that highest relative contribution to explaining the seizure power variance came from the pre-seizure HON activity (Fig. 2f).

Overall, this correlative evidence led us to hypothesize that the pre-seizure natural HON activity level may be a causal determinant of seizure intensity.

### Identifying natural HON activity epochs that control seizure intensity

To test whether pre- and during seizure epochs of natural HON activity differentially control seizure intensity, we selectively targeted these temporal epochs with inhibitory optogenetics. To achieve this, we selectively expressed the inhibitory opsin ArchT in HONs, enabling millisecond-level, reversible optosilencing of HON activity by green laser illumination[53]. We performed bilateral laser illumination of the LH, while inducing and monitoring seizures using EEG-EMG, in HON-ArchT and control mice (Fig. 3a, b). By itself, HON-ArchT optosilencing did not induce any seizure activity (Fig. 3b). Because pre-seizure HON activity positively correlated with seizure intensity (Fig. 2), we hypothesized that the HON optosilencing during this epoch would reduce seizure intensity. In support of this, we found that LH laser illumination during the pre-seizure epoch reduced seizure intensity and lowered seizure behavioral score in HON-ArchT, but not in control mice, while the same illumination during the seizure did not alter seizure electrophysiology and induced only a mild behavioral improvement (Fig. 3c, d). To probe the duration of the anti-seizure effect, we performed pre-seizure HON optosilencing, and quantifying the intensity of subsequent seizures (Fig. 3e). Using this protocol and comparing HON-ArchT and control mice, we found that the seizure-alleviating influence of HON optosilencing was short-lasting (<10 min, Fig. 3e).

Overall, these results identify pre-seizure natural HON activity as an exacerbator of seizures, and provide proof-of-concept evidence that acute suppression of natural HON activity is an effective means of suppressing seizures.

### Suppressing seizures with lateral hypothalamic deep brain stimulation

We next sought to evaluate whether LH DBS can be used in behaving mice to suppress the seizure-exacerbating HON activity. To achieve this, we implanted a DBS electrode in the LH, and monitored surrounding HONs using HON-targeted activity indicator GCaMP6s[54] and fiber photometry in our mouse seizure model (Fig. 4a). We found that sinusoid high frequency (120 Hz) hypothalamic DBS (shhDBS, based on ref. 54) significantly suppressed the seizure-implicated (i.e., pre-seizure) HON-GCaMP6s activity (Fig. 4B). In contrast, a lower stimulation frequency (2 Hz) was not effective, as expected from previous data of direct application of such frequencies to HONs in vitro[55]. In association with the suppression of HON activity, pre-seizure shhDBS significantly reduced seizure intensity, seizure probability, and seizure behavioral score (Fig. 4c, d); in contrast, seizures were not alleviated by the DBS pattern that did not suppress HON activity (Fig. 4C,D). Importantly, in our experiments, the seizure-reducing, HON-silencing shhDBS (Fig. 4c, d) did not produce side effects such as impaired movement, memory, elevated corticosterone, or body temperature[54]; nor did we ever observe any incidence of narcolepsy/cataplexy in our EEG-EMG recordings (using criteria for mouse narcolepsy/cataplexy from refs. 30,46). A further screen of side effects, including appetite and

motivation metrics as well as circadian sleep metrics, also did not reveal any DBS-induced anomalies (Supplementary Fig. 1).

To better understand the time points at which shhDBS is most effective in reducing seizures, we compared seizure intensities when shhDBS was applied pre-seizure, during seizure induction, or during the already-induced seizure (Fig. 5a, b). We found that the pre-seizure shhDBS was significantly more effective than when applied at the other times (Fig. 5c).

Finally, we asked how long the seizure-alleviating effect of shhDBS lasts. We investigated this by applying one minute of pre-seizure shhDBS, and quantifying the intensity of subsequent seizures (Fig. 5d). Using this protocol and comparing to control mice without shhDBS, we found that the seizure-protective influence of shhDBS lasted for at least 20 min after shhDBS stimulation and diminished after 24 h (Fig. 5d).

## Discussion

Current pharmacological treatments for epilepsy have substantial limitations, largely due to side effects arising from their slow and typically brain-wide modes of action[6,8]. Identification and inhibition of more specific pro-epileptic brain signals is thus a promising strategy for potential new treatments. However, even when it is targeted to restricted neural signals, chronic suppression of pro-epileptic brain activity can lead to undesirable side-effects (Fig. 1f). Therefore, an attractive alternative therapeutic logic is to identify and target temporally-defined epochs when specific neurons are particularly pro-epileptic.

The present study applied this logic to natural HON activity, using both genetically-targeted proof-of-concept tools (cell recording, optogenetic silencing) and translational, clinically-approved methods (DBS). Our real-time, concurrent detection of natural HON activity and seizures revealed substantial HON activation during epileptic seizures (Fig. 2). Surprisingly, however, we found little evidence of statistical correlation (Fig. 2f) or causal relation (Fig. 3) between this during-seizure HON activation and the seizure intensity. Instead, our correlative and causal analyses indicted that pre-seizure HON activity significantly predicted and contributed to seizure intensity (Fig. 2, Fig. 3). Temporally-restricted suppression of this pro-epileptic epoch of hypothalamic activity alleviated epilepsy in both optogenetic (Fig. 3) and shhDBS (Figs. 4, 5) experiments. Together, these results indicate that high endogenous HON activity makes the brain more vulnerable to seizure initiation, which would be consistent with known HON modulation of brain areas and physiological pathways implicated in seizures, as well as the predominantly excitatory HON outputs to much of the brain[33].

While the shhDBS and optogenetic paradigms reported here both acutely and reversibly inhibited HONs, there were interesting differences between their effects on seizure prevention. Brief optogenetic HON silencing protected from epileptic seizures only briefly (Fig. 3e), but the anti-seizure influence of similarly brief shhDBS was more prolonged (Fig. 5d). These differences could arise from several factors. Although the seizure-protective DBS inhibited pro-seizure HON activity (Fig. 4) and in vitro oscillatory stimulation can inhibit HONs directly, without stimulating their presynaptic inhibitors[55], the shhDBS is unlikely to be fully selective for HONs when applied in vivo, due to the inevitably non-selective nature of any brain electrical stimulation. The shhDBS may thus have induced other (as yet unknown) anti-seizure changes that are more long-lasting, such as changes in synaptic strengths in the HON network or elsewhere. Disentangling these effects is a key subject for future work. Irrespective of mechanism, the long-lasting nature of anti-seizure protection conferred by shhDBS is a clear translational advantage, since it increases the potential clinical benefit of the DBS as an anti-epileptic treatment. Since the DBS strategy we describe protects against seizures before they occur, it is therapeutically effective without the need for closed-loop seizure

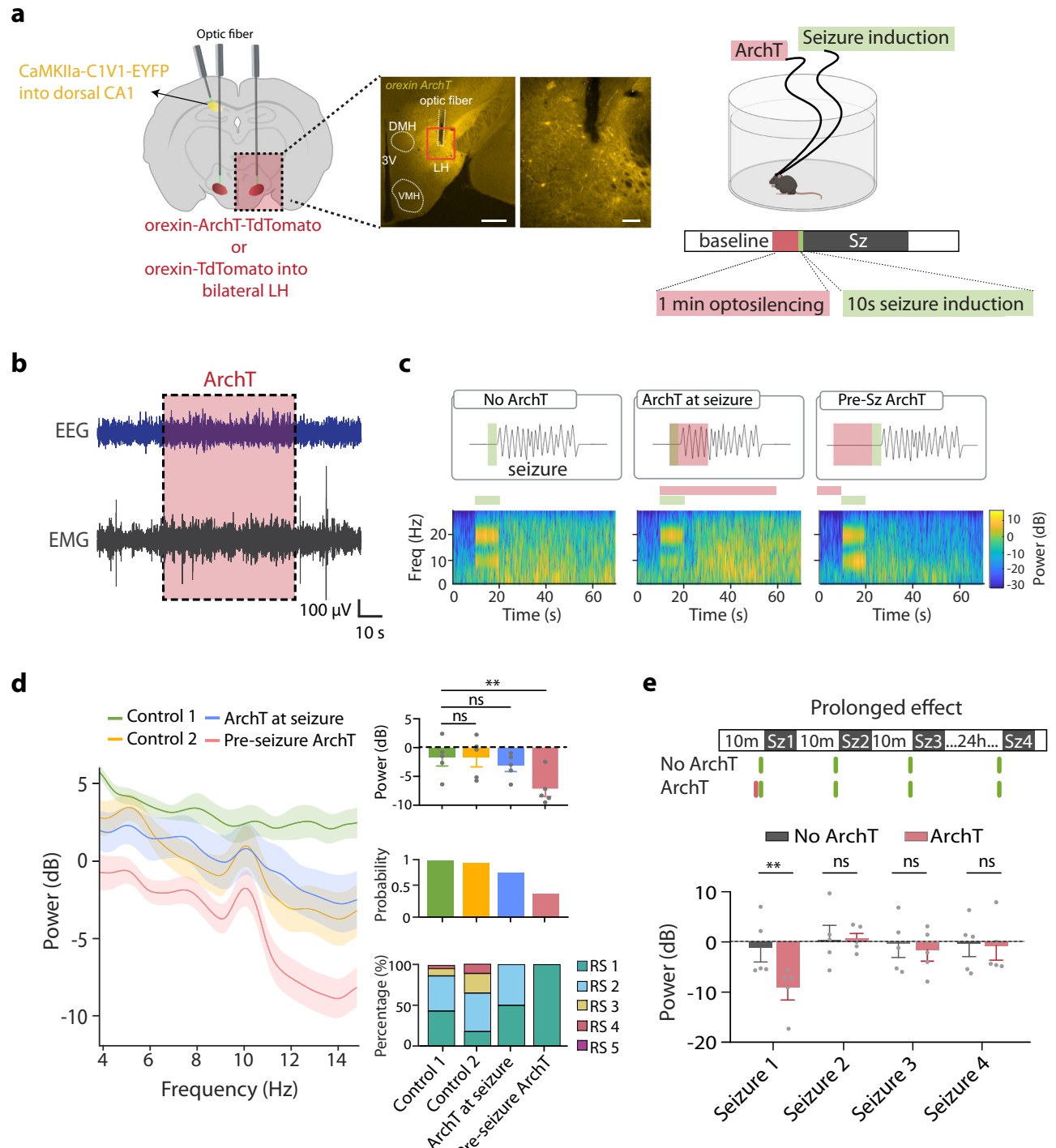

**Fig. 3 | Effect of different temporal targeting of HON optosilencing on seizures.**
**a** Left, stereotaxic surgical schematic for expression of orexin-ArchT over lateral hypothalamus. Middle, lateral hypothalamus coronal section (left, scale bar is 500 μm), showing optic fiber location and magnification (right, scale bar is 100 μm) for the red rectangular region depticted on the left. Right, experimental setup schematic (generated with BioRender), and paradigm for evaluating orexin opto-silencing effect on seizure induction. Example trial for pre-seizure optosilencing is shown. The figure is representative of *n* = 5 mice. **b** Example EEG and EMG recording showed no seizure activity during ArchT opto-silencing (dashed pink box). **c** Top, experiment protocol and Bottom, EEG time-frequency heatmap of EEG power of group data of *n* = 5 mice for ArchT inhibition (no ArchT, ArchT at seizure, and pre-seizure ArchT opto-silencing). **d** Left, spectral power analysis of seizure

EEG with no ArchT stimulation (control 1), control stimulation (control 2), ArchT during seizure, pre-seizure ArchT stimulation. Right, Top, average EEG power of the power band of interest (top, One-way ANOVA, $F_{(3,16)}$ = 3.842, *P* = 0.03; Dunnett's post-tests: control 1 vs. pre-seizure ArchT *P* = 0.02), Middle, seizure probability (middle, 98%, 95%, 66%, and 33% of 18 stimulation in each group) and Bottom, Racine score of different experimental groups. RS 1–5 as in 1 C. **e** Top, experimental protocol and Bottom, average EEG power for testing the prolonged effect (immediate vs 24 h) after ArchT or no-ArchT stimulation (for seizure 1, two-tailed *t* test *P* = 0.04; for seizure 2, 3, and 4, two-tailed *t* test *P* = 0.96, 0.72, and 0.87). Data are mean ± SEM of *n* = 5 mice. ns = not significant, **P* < 0.01. Source data are provided as a Source Data file.

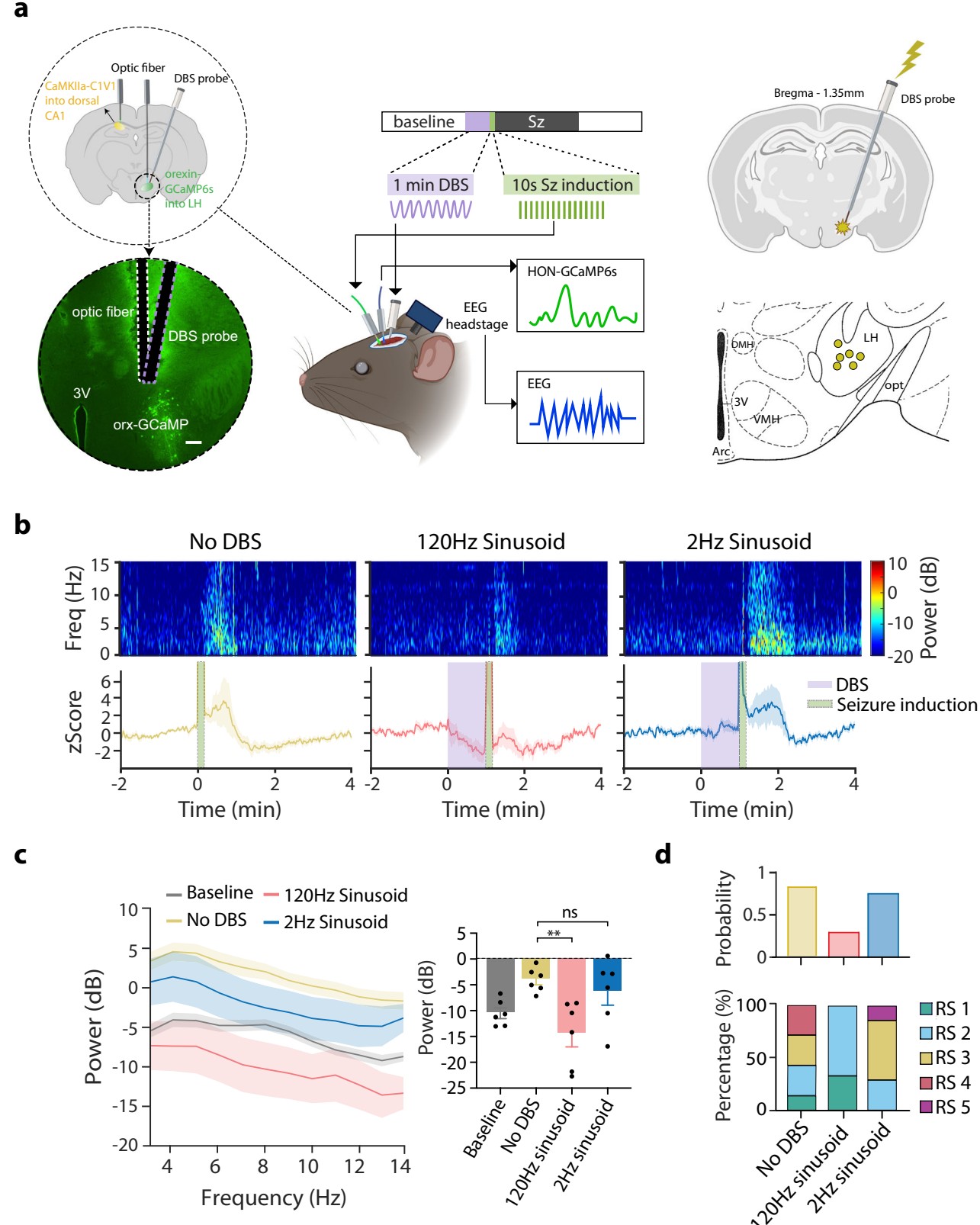

detection (Fig. 5d), which is an advantage since closed-loop demands can complicate clinical DBS applications[23,24].

Safety is a sought-after quality during the development of translational medical treatments. Our findings demonstrate that brief shhDBS leads to prolonged depression of seizures. Importantly, this DBS protocol does not cause detectable long-term motor or cognitive

impairment, and also has anxiolytic actions[54]. The prevention of seizures, without observable impact on locomotion during this DBS stimulation[54], implies little effect of shhDBS on motor control. This is an important consideration about shhDBS treatment, because strong, non-specific inhibition of the lateral hypothalamus, for example by targeted electrolytic lesions, can dramatically reduce general

**Fig. 4 | Effect on pro-seizure HON activity of different protocols of lateral hypothalamic DBS. a** Left, schematic for virus expression and locations of fiber and DBS probe placement (top) and lateral hypothalamus coronal section, scale bar 200 μm (bottom). Middle, the experimental paradigm for fiber photometry and EEG recordings during DBS and seizure induction. Right, diagrams of DBS probe placement of all 6 mice in the below experiments. Schematic cartoons generated with BioRender. **b** Top, time-frequency heatmap of EEG power of group data of $n = 6$ mice; Bottom, HON photometry for different DBS stimulation patterns during seizure induction. **c** Left, spectral power analysis, and Right, average EEG power between groups of baseline (gray), no DBS (yellow), 120 Hz sinusoidal DBS (red), and 2 Hz sinusoidal DBS (blue) (One-way ANOVA, $F_{(3,20)} = 5.382$, $P = 0.007$; Dunnett's post-tests: No DBS vs.120 Hz DBS $P = 0.004$). **d** Top, seizure probability (89%, 29%, and 75% of 15 stimulation in each group) and Bottom, Racine score of different experimental groups. RS 1–5 as in 1 C. Data are mean ± SEM of $n = 6$ mice. ns = not significant, **$P < 0.01$. Source data are provided as a Source Data file.

motivation and locomotion[56]. While we found that shhDBS suppressed an element of lateral hypothalamic activation, consistent with its anti-seizure properties, this suppression was presumably not strong enough to cause the general impairment observed upon lateral hypothalamic lesions. Importantly, our anti-seizure shhDBS did not itself produce detectable epileptiform brain activity, a central concern in DBS treatments[57,58]. In fact, we did not observe any side effects of shhDBS, having previously screened for diverse parameters, including effects on movement, memory, stress hormones, and body temperature[54], as well as additional parameters in the present study (Supplementary Fig. 1).

Our findings suggest important directions for future work. The electrically excitatory outputs of HONs are thought to be mediated by the release of orexin peptides acting on two types of orexin G-protein coupled receptors[29,33,59] (both of which are thought to be antagonized by the concentration of SB we used in Fig. 1[60]), as well as by the excitatory neurotransmitter glutamate that is co-released from HONs[34,61–63]. This suggests that multiple excitatory transmitters may contribute to pro-seizure effects of HONs, and elucidating their relative roles is a key subject for further investigation. Furthermore, the focal seizure we used is intended to model hippocampus-originating human epilepsy[48], but many clinical seizures involve a broader network of structures and cause different epilepsy syndromes[64]. It would therefore be important to investigate whether the effectiveness of the hypothalamic interventions described here also extends to other seizure models, such as neocortical, generalized, and genetic seizures[65], as well as to females since only males were used in the current study. Although so far we did not observe adverse side effects of LH DBS in a variety of paradigms (Supplementary Fig. 1), further long-terms effects in a variety of seizure models remain to be investigated.

Overall, our results thus suggest that lateral hypothalamic DBS could be a promising strategy to augment treatment for epilepsy, to help to prevent seizures. DBS targeting of the hypothalamus is feasible in human patients[66], in principle enabling future examinations of whether certain lateral hypothalamic DBS patterns could be effective in human epilepsy. In contrast to common epilepsy treatments, such as the medications that have non-specific brain-wide effects, DBS has the potential to modulate the activity of a brain region that is currently not widely targeted in therapies of this disorder. Such a targeted approach may improve therapeutic efficacy, reduce side effects, and provide a treatment alternative for epilepsy patients that are resistant to current therapies. Furthermore, since shhDBS is anxiolytic[54], it may also alleviate psychopathological symptoms (such as stress and fear) associated with seizures[67].

## Methods
### Genetic and anatomical targeting
Subjects were adult male C57BL/6 mice, over the age of eight weeks. They were kept on standard chow and water ad libitum, and housed in a reversed 12/12 h dark/light cycle, in a temperature ($21 \pm 1\,°C$) and humidity ($50 \pm 5\%$) controlled facility. The experiments were performed during the dark phase. To express genetically targeted activity indicators and controllers in HONs, we used orexin promotor (hORX) vectors that were developed and histologically validated to be specific for HONs in our previous studies[52,53,68–70]. Briefly,

to target GCaMP6s to orexin neurons, we injected AAV1-hORX.GCaMP6s (titer: $2.0 \times 10^{13}$ GC/ml; prepared by Penn Vector Core) into the LH (at coordinates given below). To target the opto-genetic inhibitory opsin ArchT to orexin neurons, we injected AAV1-hORX.ArchT.TdTomato (1:20 dilution, titer: $1.03 \times 10^{13}$ GC/ml; prepared by Vigene Biosciences) into the LH. For the virus control in optogenetic experiments, we injected hORX.TdTomato as a non-opsin control. For induction of epileptic seizures, we injected AAV9-CaMKIIa.C1V1(t/t).TS.EYFP (titer: $2.3 \times 10^{13}$ GC/ml; prepared by Addgene, MA, USA) into the unilateral dorsal hippocampus[71].

For stereotaxic brain surgeries, isoflurane anaesthesia was maintained (1% at 1 l/min) and a single dose of Metacam (5 mg/kg) was injected subcutaneously as local analgesia. Mice were then placed in a stereotaxic frame (Kopf Instruments, Tujunga, USA) with a heating pad to maintain body temperature. For long surgeries, phosphate-buffered saline was injected subcutaneously to prevent dehydration. Chlorhexidine (2%) was used for wound cleaning, and a craniotomy was performed in the midline. After connective tissues were removed, bregma and lambda were aligned, and holes were drilled at the below coordinates relative to bregma. A Hamilton syringe was used for viral vector injection at a rate of 50 nl/min. For AAV vectors targeting orexin neurons, 150–250 nl of GCaMP6s AAV vector or 400 nl of ArchT AAV vector was injected into the LH (bregma, −1.35 mm; midline, ±0.95 mm; depth, 5.40 mm). For induction of seizure, 150 nl of CaMKIIa.C1V1 virus was injected into the dorsal hippocampus (bregma, −2.06 mm; midline, ±1.30 mm; depth, 1.25 mm).

For experiments with seizure induction and fiber photometry, fiberoptics (CFML12U-20, 200 μm in diameter, 0.39NA, Thorlabs, New Jersey, USA) were stereotaxically implanted with the fiber tip 500 μm above the target region. For experiments combined with DBS stimulation, a bipolar concentric stimulation electrode (P1 Technologies, Roanoke VA, USA, custom 1-mm contact separation) was implanted unilaterally on the same LH side as the GCaMP6s fiberoptic at bregma, −1.35 mm; midline, ±2.70 mm; depth, 5.10–5.20 mm, with a 20-degree angle. For the electroencephalogram (EEG) and electromyogram (EMG) surgery, a 3-channel EEG/EMG headmount (Pinnacle Technology, 8235: 6-Pin Headmount, USA) with four stainless steel screws that acted as cortical surface electrodes were implanted into the skill. Briefly, anterior screws were 1 mm anterior to bregma and 1 mm lateral from the midline on each side, and posterior screws were 4.5 mm posterior to bregma and 2 mm lateral from the midline. The left anterior screw was used as the ground connection. A pair of stainless steel wires were embedded into the posterior neck muscle to record EMG signals (Fig. 2a). All the implanted probes and headmounts were attached and fixed to the skull with dental cement (Super-bond universal kit, Hentschel-Dental, Teningen/OT-Nimburg, Germany). Mice recovered from surgery for at least 10 d before experiments.

### EEG-EMG recordings
For the EEG-EMG recordings (Fig. 2b), data were acquired at a 1-kHz sample rate by a standard acquisition system with a commercial pre-amplifier (Pinnacle Technology, Lawrence, KS, USA; 10x gain). Data were recorded with Spike2 software via a CED Micro1401-3 data acquisition unit (Cambridge Electronic Design, Cambridge, UK). Raw EEG signals were first filtered with a low-pass filter at 100 Hz to remove DBS stimulation artefacts, and a second-order 50 Hz notch filter to

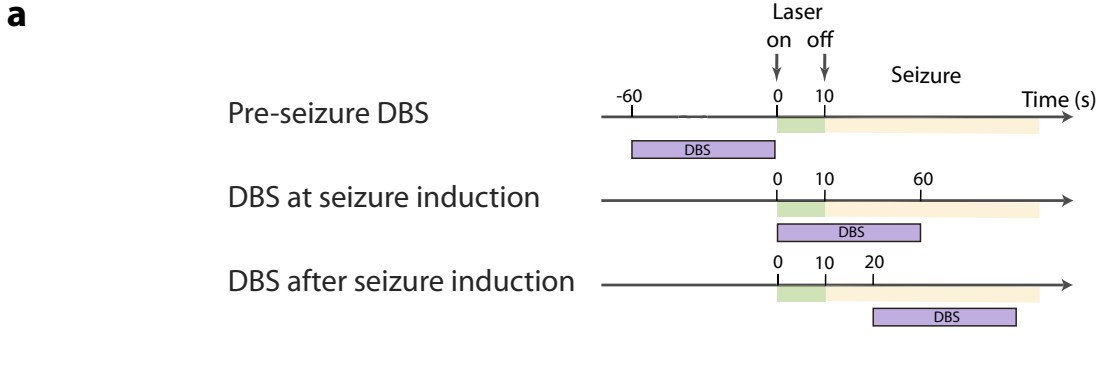

**Fig. 5 | Temporal properties of the effect of lateral hypothalamic DBS on seizures. a** Experimental protocol for DBS at different time points during seizure induction. **b** Example EEG recording demonstrated pre-seizure DBS, DBS at seizure induction, and DBS after seizure induction. The dashed purple square box indicated the DBS stimulation point. **c** Average EEG power in groups of no DBS, pre-seizure DBS, DBS with seizure induction, and DBS after seizure induction (One-way ANOVA, $F_{(3,20)} = 6.936$, $P = 0.002$; Tukey's post-tests: No DBS vs. pre-seizure DBS

$P = 0.01$; pre-seizure DBS vs. DBS after seizure induction $P = 0.004$). Data are mean ± SEM of $n = 6$ mice. **d** Top, experimental protocol; Bottom, average EEG power for testing the prolonged effect (immediate vs 24 h) after DBS or no-DBS stimulation (for seizure 1, 2, and 3, two-tailed $t$ test $P = 0.01$, 0,02 and 0.003; for seizure 4, two-tailed $t$ test $P = 0.91$). Data are mean ± SEM of $n = $ DBS/control = 3/3 mice. ns = not significant, \*$P < 0.05$, \*\*$P < 0.01$. Source data are provided as a Source Data file.

remove line noise. For the EEG power spectral analysis, the Welch method was used to compute the mean power spectral density estimate of signals which were segmented into 1-s epochs. The average band power was calculated at the specific frequency of 4–14 Hz

(Fig. 1b). The signal duration for analysis was set for 30 s after seizure induction, referenced from the average seizure duration indicated in Fig. 1d. We performed all signal analyses as described using MATLAB (MathWorks, USA).

## Neural recordings

Fiber photometry of HON-GCaMP6s activity was recorded using the interleaved mode of the Doric fiber photometry system (405- and 465-nm excitation light pulses, interleaved at 10 Hz, with an excitation light power of ≈100 µW at the fiber tip)[54,72]. Data were recorded using Spike2 software at a sampling rate of 1-kHz. For each recording session, a linear fit function was fitted to correct linear bleaching of the signals (raw 405- and 465- fluorescence trace was first detrended through the median first and last 50 s of 6-minute time window centered on the time point of seizure induction). Next, for each peri-stimulation time epoch, we did a z-score normalization for the fitted traces using median and standard deviation of the trace baseline 60 s before seizure induction. Finally, to correct for movement artefacts, the z-scored 405-excitation trace was subtracted from 465-excitation trace because the 405-excitation is an isosbestic point of GCaMP6s which reports movement artefacts but not neural $Ca^{2+}$ activity[73].

## Optogenetics

For hippocampal induction of epileptic seizures, we stimulated dorsal hippocampal CaMKIIa-C1V1-expressing neurons using a 532 nm laser (Laserglow, ≈10 mW at the optic fiber tip) for 10 seconds, with a 5 ms pulse duration, at frequencies of 5, 10, or 20 Hz, as stated in the figure legends (green bar in Fig. 1a). TTL output from the laser was recorded in Spike2 software and was used to align with EEG and photometry data for further analysis. To validate that the seizure was induced successfully, we observed the seizure-like after-discharges during the ictal phase and calculated the whole stage as seizure duration (Fig. 1a). To evaluate the behavioral score for each stimulation session, we recorded the video of the experiment with an infrared camera and later used automated video tracking software EthoVision XT (Noldus, Wageningen, the Netherlands) for analysis. Behavioral Racine score of seizures was manually scored based on the following observations from video: RS1 - sudden arrest, mouth, and facial movement; RS2 - head nodding and stiffened tail; RS3 - body myoclonic jerks with lordotic posture; RS4 - rearing with forelimb clonus; RS5 - tonic-clonic seizures with loss of posture[51]. For the calculation of seizure probability, we counted all provoked seizures, divided by numbers of all stimulation for all mice (as described in figure legends per experiment).

For optogenetic manipulation of HONs in Fig. 3, bilateral optogenetic stimulation of HON-ArchT cells was performed via LH fiberoptics using 532 nm green lasers (5 ms flash duration, 10 Hz flash frequency, ≈10 mW laser power at fiber tip), as in ref. 53. For the control experiment, either no ArchT stimulation (Control 1) or sham stimulation with hORX.TdTomato (Control 2) with similar laser stimulation protocols was applied.

## Experimental DBS protocol

We programmed the DBS stimulation protocol with Patchmaster software (Heka Elektronik, Lambrecht Germany), and operated it through an amplifier (HEKA EPC-10). A bipolar cable (P1 Technologies, Roanoke VA, USA) was connected from the amplifier to the DBS electrode to deliver continuous electric stimulus. The sinusoidal oscillation waveforms were delivered at frequencies of either 2 or 120 Hz as in ref. 54. Classic (non-sinusoid) DBS was applied as in refs. 54,74. The peak-to-peak voltage was 2.0–4.0 V tested in an escalating fashion for each mouse, to achieve a consistent HON inhibition without seizure induction or other behavioral abnormalities. The stimulation duration was 1-min pre-, at, or after seizure induction as described in figure legends per experiment. The control group (Fig. 5d) had no DBS stimulation during the testing of the prolonged effect, as described in the optogenetic experiment.

## Behavioral tests

**Sucrose preference test (SPT).** The sucrose preference test utilizes a two-bottle choice paradigm to measure anhedonia. For this test, an experimental cage resembling the animal's home cage was set up with two bottles: one filled with plain drinking water and the other with a 3% sucrose solution (Sucrose, Sigma-Aldrich). No food was present during the test. One day before the actual test, animals were acclimated to the cage equipped with two water bottles. Additionally, they were familiarized with the sucrose solution in their home cage. Before the test commenced, animals underwent mild food and water deprivation. As a baseline reference, the SPT was conducted without DBS on day 0. On subsequent days (days 1–3), the two original bottles were replaced: one with water and the other with the 3% sucrose solution. The 120 Hz DBS was initiated at the beginning of the test. To track liquid consumption during the 1-h test, both bottles were weighed before and after placement in the cage. 24 h after the final SPT, animals were retested using the SPT setup, but without the DBS intervention. To mitigate potential side bias, the positions of the bottles were alternated daily. The sucrose preference index was determined by dividing the volume of sucrose consumed during the test by the total volume of liquid consumed.

**Forced swim test (FST).** The forced swim test was used to evaluate "depressive-like" states and behavioral despair. The test was conducted in a plastic cylinder measuring 25 cm in height and 20 cm in diameter. Prior to the test, the cylinder was filled to a depth of 15 cm with water maintained at 23 °C. Animals were then introduced into the cylinder, allowing them to swim for a duration of 5 min. As a baseline reference, the FST was conducted without DBS on day 0. On day 1, DBS was administered for 5 min within the animal's home cage. Immediately after the stimulation concluded, the animal was placed into the FST cylinder. The entire process was captured on video and subsequently analyzed using the Ethovision software. Two primary behaviors were recorded: immobility, indicative of passive coping, and swimming, reflecting active coping. Immobility was characterized as minimal movement, primarily small efforts to keep the head above water. In contrast, swimming was detected when the animal moved around the cylinder with all four paws underwater.

For the analysis the cumulative duration of the last four minutes was taken, and the results for the two behavioral states were reported in seconds.

**Appetite test.** To evaluate potential changes in general appetite and spontaneous locomotor activity post-DBS administration, we designed an experimental setup to monitor both standard chow consumption and movement within the home cage. On day 0, the baseline was established for chow intake over a 3-hour span and distance moved was captured using video recording. From days 1–3, following a 1-h DBS session, chow consumption was monitored for the subsequent 3 h. The total distance covered during the initial 40 min of the DBS session was recorded using Ethovision software.

**Motivation test.** To assess the impact of DBS administration on reward-related behavior, we employed a motivation test. Over the initial two days, mice were familiarized with both the experimental chamber, measuring 60 cm in length and 40 cm in width, and chocolate-flavored pellets (Bio-Serv) which were provided in their home cage for a duration of 30 minutes. The testing environment was illuminated with white light at an intensity of 40 lux. On day 2, animals were permitted to explore the experimental chamber for a 10-min duration. During this time, they had unfettered access to two different trays: one containing standard chow and the other stocked with chocolate pellets. The trays were placed a specified distance (15 cm) apart. On day 3, the test was repeated, but with the addition of DBS administration throughout the test's duration. All test sessions were video-recorded and subsequently analyzed using the Ethovision software. To estimate motivation, we measured the time in seconds taken to first

approach and first consume the food items, along with the total consumption of both food types (standard chow vs. chocolate pellets).

**Sleep EEG recording setup, experimental protocols and analysis**
After EEG and DBS implantation surgery, animals recovered in a reversed light cycle room for at least 2 weeks. Animals were then habituated to the EEG recording chamber for 3 days, which was equipped with nesting bedding, water bottle, food chow, infrared camera, and a lamp with a timer controller (7 pm: light on, 7am: light off). After tether-habituation, we conducted continuous EEG/EMG/video recordings for 2 days to establish a baseline reference. To assess long-term changes in sleep patterns after chronic DBS stimulation, we arranged the behavioral experiments (SPT and appetite test) during a 2-week period (a total of six 1-hr stimulation sessions for each mouse).

Next, we started the sleep EEG recording plus DBS stimulation during different sleep states. On the first day, mice received 1-h DBS stimulation at 4 h after dark phase onset (ZT16-17). On the second day, following a >24-h washout period, mice received same stimulation at 4 h after light phase onset (ZT4-5). We then reviewed the raw EEG data and scored three sleeping states based on EEG/EMG patterns: wakefulness (Wake), NREM sleep and REM sleep. All scored sessions were assessed and expressed as percentage time spent in different sleep states during either dark or light phase. To measure the sleep fragmentation index, we first defined the behavioral bout as a series of 5 s epochs without transitions. Next, we calculated the resulting number of behavioral bouts, divided by the total epochs in the same stage, leading to a comparable fragmentation index between 0 and 1[75].

**Histology**
After being sedated with pentobarbital i.p injection, mice were transcardially perfused with PBS followed by 4% PFA. The brain implants and skull bones were carefully removed, and brains were post-fixed for 24 h and transferred to phosphate-buffered saline for long-term storage. Coronal brain slices were sectioned at 50 μm using a cryostat. Micrographs were acquired by Nikon Eclipse Ti2 and the resultant image stacks were processed in an image analysis program (Fiji)[76].

**Statistics**
Raw data were processed in MATLAB 2019b (Mathworks, USA). Statistical analysis was done using Prism 9 (GraphPad Software Inc, California, USA). Data were assessed for normality with a D'Agostino–Pearson omnibus test or Kolmogorov–Smirnov test before parametric tests. *T*-tests were used to compare experimental data of two groups, and one-way repeated measures (RM) ANOVA was used for comparing experiments of multiple groups and with multiple comparison tests where appropriate. Data were presented as mean ± SEM in the result and figures. *P* values < 0.05 were considered statistically significant. The relative contributions of HON responses during different seizure phases to EEG power of seizure (Fig. 2) were determined using a generalized linear model approach[77]. In the model, EEG power was used as the response variable, whereas predictor variables were HON activity 60 s pre- and during seizure induction (Fig. 2e). A total of 30 recordings collected from 8 mice were used for the data analyses and model development. Here 70% of the data was used for training the model and 30% was used as test data to determine the relative influence of the predictor variables. Specifically, we used the "glmfit" function in MATLAB to fit the data by bootstrapping training data randomly over 5000 iterations. Using this model a coefficient of determination ($R^2$) was calculated for the validation data. To determine the relative contribution of a given predictor to EEG power, we quantified how much the encoding model performance declined without the given variable. Negative contributions were set to zero (Fig. 2f).

**Study approval**
All animal experiments were performed according to the Swiss Federal Food Safety and Veterinary Office Welfare Ordinance (TSchV 455.1, approved by the Zurich Cantonal Veterinary Office), or with the UK Home Office regulations.

**Reporting summary**
Further information on research design is available in the Nature Portfolio Reporting Summary linked to this article.

## Data availability
The data generated in this study have been deposited in the Figshare repository (https://doi.org/10.6084/m9.figshare.24167898)[78]. Source data are provided with this paper.

## Code availability
The code to analyze the preprocessed data are available at Figshare repository (https://doi.org/10.6084/m9.figshare.24167898)[78]. Further code is available upon request.

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

## Acknowledgements

This work is funded by ETH Zürich and Chang Gung Memorial Hospital, Taiwan (Grant CMRPG3N0261). We thank Dr. Lukas Imbach, Dr. Dane Donegan and Prof. Mahesh Karnani for constructive comments on the manuscript. Cartoons in Fig. 1a, 2a, b, 3a, 4a, and Supplementary Fig. 1 were created with BioRender.com.

## Author contributions

HTL obtained funding, designed and carried out experiments and contributed to writing; PV assisted in mathematical analyses; EB assisted in experiments; DPR designed and coordinated the behavioral experiments probing side effects and also contributed to the overall study strategy; DB obtained funding, contributed to experimental design, and wrote the paper with input from all co-authors.

## Funding

## Competing interests

The authors declare no competing interests.
