## [Peer Review File · Nature Communications]

Transient targeting of hypothalamic orexin neurons alleviates seizures in a mouse model of epilepsyREVIEWER COMMENTS

Reviewer #1 (Remarks to the Author):

Review Nature NCOMMS-23-16468 orexin and seizures

Disclaimer: My expertise is in clinical and animal models of epilepsy and of neurostimulation, but not optogenetics, so others will have to speak to the optogenetic aspects. From an epilepsy standpoint, this is a very interesting paper with information new to the epilepsy community. Experiments and controls appear to be carefully constructed.

Seizures were produced with optogenetic stimulation of mouse hippocampus. Lateral hypothalamic hypocretin/orexin neurons (HON) activity fluctuates. HON activity before triggering hippocampal seizures optogenetically significantly correlated with seizure intensity. The inhibitory opsin ArchT in HONs, reversibly inhibited activity over milliseconds and could be used to produce seizure-suppression. This effect was replicated, over a longer time-course, by high-frequency lateral hypothalamic stimulation.

SPECIFICITY: Please discuss the specificity of the orexin antagonist, SB-334867. Does the anti-seizure effect depend upon OX-1 receptors or is OX-2 also involved?

BETTER EXPLAIN HOW SEIZURE INTENSITY WAS MEASURED: Using the behavioral Racine scale is clear, but what about the EEG? Total spectral power? Power in a specific band? Duration?

APPLICABILITY OF THE MODEL: Optogenetically-triggered hippocampal seizures provide a good model in terms of controllability in time and place. But most clinical seizures involve a broad network and structures outside hippocampus. Authors should discuss model-dependence of their findings. Would the same results occur with neocortical seizures, generalized seizures from the PTZ or lithium-pilocarpine, kainite or genetic epilepsy models? I am obviously not asking that they do these experiments, but only mention the limits of extrapolating from one very specific epilepsy model in one species.

SIDE EFFECTS: Side effects do not mention appetite changes, which is thought to be a key function of lateral hypothalamus. There is surprisingly little discussion of effects on sleep, given HON key role in promoting wakefulness. It is well-known among clinicians that focal onset seizures are more frequent during drowsiness and light sleep. Does that proclivity relate to this system? Can their findings potentially explain this relationship?

EFFECT DURING SEIZURES: Figure 3D seems to show that ArchT during the seizure inhibits the Racine stage from progressing beyond RS2. How does that fit with the discussion about how seizure inhibition by HON must be before the seizure?

CLINICAL UTILITY: Assuming that activation of HON is only effective before a seizure, this could limit clinical utility. Since the seizure-inhibitory effect lasts 10 minutes, perhaps the common cyclical 5-minute DBS stimulation cycle could be effective. But this would raise the issue of advisability of permanently affecting this system.

Reviewer #2 (Remarks to the Author):

This is a well-conducted study of great translational importance. The finding that HONs contribute to seizure activity is quite novel and brings forth a lot of exciting new questions about the role of these neurons in regulation of brain-wide activity states. This paper is exciting because it provides a cell-type specific mechanism of a very broad manipulation (shhDBS in LH). My major concerns regard the

potential side effects of the shhDBS, which I think should be explored further.

-Strengths:

- o They clearly show that HON activity preceding seizures causally contributes to seizure intensity. The time-dependence of this effect is really interesting.
- o The effect of DBS appears reversible, which is important if the protocol is to prevent long-lasting behavioral or mood changes in clinical populations.

- Weaknesses:

- o No use of female subjects, or even mention of them. The authors should at least mention a lack of female subjects as a major weakness in their study.
- o Authors should do more controls related to the specificity of this effect on seizures only. Even if animals aren't going to sleep, it is possible that the animals are anhedonic, show low motivation, changes in coping strategies, longer-term changes in sleep patterns etc. Other behavioral measures related to motivation should be tested given that the PNAS paper shows an effect of shhDBS on RTP. Examples of tests the authors can do are sucrose preference tests, EEG sleep recordings, and the forced swim test (active vs passive coping).
- o It is important to test what the effects of repeated DBS are on a variety of locomotor, reward-related, and mood-related behaviors, as this is really critical for translatability of these findings. This is especially true because the authors state that shhDBS could be a strategy to prevent seizures before they occur, which means that the shhDBS needs to be on much of the time, not just for short time epochs preceding induced seizures.
- o Authors should provide dose given for SB compound
- o The scope images of the representative infections are of very poor quality. The viral expression for Arch is especially sparse and looks far too medial for HON neurons. The authors should provide diagrams of DBS probe placement and viral infections for these experiments. It concerns me that that this Arch image was the best the authors had to show as a "representative" image. Please show other images so we can get a sense of whether there was sufficient expression of the opsin in HONs.
- o In figure 5B it would be helpful to also see a trace of what DBS following seizure induction looks like.

NCOMMS-23-16468, Authors' response to reviewers.

We are grateful to the Reviewers for their kind comments and constructive suggestions, that have improved our manuscript. Below, their comments are in italics and our responses are in bold. The changes to the text are highlighted in blue in the revised manuscript.

Reviewer #1:

From an epilepsy standpoint, this is a very interesting paper with information new to the epilepsy community. Experiments and controls appear to be carefully constructed. Thank you very much for these supportive comments. Seizures were produced with optogenetic stimulation of mouse hippocampus. Lateral hypothalamic hypocretin/orexin neurons (HON) activity fluctuates. HON activity before triggering hippocampal seizures optogenetically significantly correlated with seizure intensity. The inhibitory opsin ArchT in HONs, reversibly inhibited activity over milliseconds and could be used to produce seizure-suppression. This effect was replicated, over a longer time-course, by high-frequency lateral hypothalamic stimulation.

SPECIFICITY: Please discuss the specificity of the orexin antagonist, SB-334867. Does the anti-seizure effect depend upon OX-1 receptors or is OX-2 also involved?

Both of the orexin receptors are neuroexcitatory, which would be expected to contribute to seizures. At the concentrations typically used in in vivo experiments, including our study, SB is believed to oppose orexin binding to both receptors. Interestingly, HONs also release the excitatory transmitter glutamate, which may further contribute to seizures. Thus, anti-seizure effects of HON suppression likely involve multiple receptors and transmitters. We have added a discussion of this point on (p. 10, first half of first paragraph), as requested.

BETTER EXPLAIN HOW SEIZURE INTENSITY WAS MEASURED: Using the behavioral Racine scale is clear, but what about the EEG? Total spectral power? Power in a specific band? Duration?

For the EEG power spectral analysis, the Welch method was used to compute the mean power spectral density estimate of signals which were segmented into 1-s epochs. The average band power was calculated at the specific frequency of 4 – 14 Hz (Fig. 1B). The signal duration for analysis was set for 30 seconds after seizure induction, referenced from the average seizure duration indicated in figure 1D. This is explained on p.12-13, as requested.

APPLICABILITY OF THE MODEL: Optogenetically-triggered hippocampal seizures provide a good model in terms of controllability in time and place. But most clinical seizures involve a broad network and structures outside hippocampus. Authors should discuss model-dependence of their findings. Would the same results occur with neocortical seizures, generalized seizures from the PTZ or lithium-pilocarpine, kainite or genetic epilepsy models? I am obviously not asking that they do these experiments, but only mention the limits of extrapolating from one very specific epilepsy model in one species.

We agree with the Reviewer, and now added a comment on these limits in the Discussion (p. 10): “the focal seizure we used is intended to model hippocampus-originating human epilepsy, but many clinical seizures involve a broader network of structures and cause different epilepsy syndromes. It would therefore be important to investigate whether the effectiveness of the hypothalamic interventions described here also extends to other seizure models, such as neocortical, generalized, and genetic seizure.”

SIDE EFFECTS: Side effects do not mention appetite changes, which is thought to be a key function of lateral hypothalamus. There is surprisingly little discussion of effects on sleep, given HON key role in promoting wakefulness.

We have now added new data on these important points (Supplementary Figure 1). These new results suggest that our DBS does not cause undesirable side-effects in these parameters (data shown and described in our reply to Reviewer 2, point 2).

It is well-known among clinicians that focal onset seizures are more frequent during drowsiness and light sleep. Does that proclivity relate to this system? Can their findings potentially explain this relationship?

To the best of our knowledge, the activity level of HONs during drowsiness and light sleep in humans is currently unknown. Therefore, we unfortunately cannot answer this directly.

EFFECT DURING SEIZURES: Figure 3D seems to show that ArchT during the seizure inhibits the Racine stage from progressing beyond RS2. How does that fit with the discussion about how seizure inhibition by HON must be before the seizure?

It is true that there is a slight improvement in this behavioral parameter, however, it is smaller than that with “pre-seizure” stimulation. There is no improvement in electrophysiological parameters with during-seizure stimulation. Thus our main conclusions still hold, but we now added more detail to text (p. 5) “LH laser illumination during the pre-seizure epoch reduced seizure intensity and lowered seizure behavioral score in HON-ArchT, but not in control mice, while the same illumination during the seizure was ineffective at the electrophysiological level and induced only a mild behavioral improvement”.

CLINICAL UTILITY: Assuming that activation of HON is only effective before a seizure, this could limit clinical utility. Since the seizure-inhibitory effect lasts 10 minutes, perhaps the common cyclical 5- minute DBS stimulation cycle could be effective. But this would raise the issue of advisability of permanently affecting this system.

We agree that these are important points. The balance between efficacy and battery consumption is a critical issue for DBS. For conventional anterior thalamic nucleus stimulation for epilepsy, the cycling-mode was usually set at a frequency around 90-140Hz, with cycles of 1-minute on and 5-minute off. Based on our findings, the LH DBS has a seizure protective effect of around 10 minutes (Fig 5D); therefore, using the conventional cycling-mode should be reasonable and effective. Regarding the permanent effects of the system, based on our new behavioral data (please see our response to Reviewer 2 point 2), there were no obvious signs of HON system-related physiological changes (appetite, locomotion, reward seeking, sleep) after chronic stimulation. We now did this over one month in the mouse model (i.e. 5-8% of its lifespan. Further long-term effects remained to be investigated. We now added a Discussion of these points on p. 10.

Reviewer #2 (Remarks to the Author):

This is a well-conducted study of great translational importance. The finding that HONs contribute to seizure activity is quite novel and brings forth a lot of exciting new questions about the role of these neurons in regulation of brain-wide activity states. This paper is exciting because it provides a cell-type specific mechanism of a very broad manipulation (shhDBS in LH). My major concerns regard the potential side effects of the shhDBS, which I think should be explored further.

-Strengths:

- o They clearly show that HON activity preceding seizures causally contributes to seizure intensity. The time-dependence of this effect is really interesting.*
- o The effect of DBS appears reversible, which is important if the protocol is to prevent long-lasting behavioral or mood changes in clinical populations.*

Thank you very much for these supportive comments. We now include several additional datasets in the revised study, that further explore side effects.

- Weaknesses:

- 1. No use of female subjects, or even mention of them. The authors should at least mention a lack of female subjects as a major weakness in their study.*

Apologies for this oversight. The present study used only male mice, we now state this more clearly (in the abstract), and mention that it would be important to extend the investigation to females in the future (p. 10).

- 2. Authors should do more controls related to the specificity of this effect on seizures only. Even if animals aren't going to sleep, it is possible that the animals are anhedonic, show low motivation, changes in coping strategies, longer-term changes in sleep patterns etc. Other behavioral measures related to motivation should be tested given that the PNAS paper shows an effect of shhDBS on RTP. Examples of tests the authors can do are sucrose preference tests, EEG sleep recordings, and the forced swim test (active vs passive coping).*

- 3. It is important to test what the effects of repeated DBS are on a variety of locomotor, reward-related, and mood-related behaviors, as this is really critical for translatability of these findings. This is especially true because the authors state that shhDBS could be a strategy to prevent seizures before they occur, which means that the shhDBS needs to be on much of the time, not just for short time epochs preceding induced seizures.*

We agree that these are very important points. To address them, we now carried out an extensive additional set of experiments (please see data below). In these new experiments, we did not observe effects on hedonic/appetite state (sucrose preference test, appetite test), assessing depressive-like behaviors such as coping strategies (forced swim test) or anhedonia, the ability to experience pleasure (sucrose preference test), reward-responsiveness the motivation to approach a possible reward spontaneously (motivation test), and sleep patterns (assayed here as day-night sleep state quantification and sleep fragmentation quantification). Within these experiments, to assess any behavioural or sleep pattern changes after repeated DBS stimulation, we designed different behavioural paradigms with a longer DBS duration, from 5-min (forced swim test), 10-min (motivation test) to 1-hr (sucrose preference test, appetite test, and sleep recording) per stimulation session as well as chronic effects of DBS application over several days. No significant changes related to HON were observed in physiology or behavior during DBS sessions or 24 hours afterward when DBS was not applied. These new data thus further support our original conclusions regarding specificity, we thank the Reviewer for suggesting these additional tests.

Requested additional probing of behavioural, affective, and brain-state side-effects (this has now been added to the manuscript as new Supplementary Figure 1, and described on p. 6 of Results and in the new Methods):

4. Authors should provide dose given for SB compound
Now added on p. 4.

5. The scope images of the representative infections are of very poor quality. The viral expression for Arch is especially sparse and looks far too medial for HON neurons. The authors should provide diagrams of DBS probe placement and viral infections for these experiments. It concerns me that that this Arch image was the best the authors had to show as a “representative” image. Please show other images so we can get a sense of whether there was sufficient expression of the opsin in HONs.
As requested, we have now added new histology plots (Fig. 3A) and a further diagram of DBS placement (Fig. 4A).

6. In figure 5B it would be helpful to also see a trace of what DBS following seizure induction looks like.
We have now added this as requested.

REVIEWERS' COMMENTS

Reviewer #1 (Remarks to the Author):

All of my suggestions have been addressed and this excellent study is now even stronger.

Robert S. Fisher, MD, PhD, Stanford

Reviewer #2 (Remarks to the Author):

The authors have addressed all of my concerns and the manuscript is much improved. The authors should be proud of such a translationally important piece of work.

But, please include female mice next time! There was no need to exclude them in this study (and I think the authors knew that). Because of this, we now only have the benefits of knowing this really useful information in men/males. Do better!

The Reviewers were satisfied and did not ask for any further revisions.